# Accuracy of the Geriatric Depression Scale (GDS)-4 and GDS-5 for the screening of depression among older adults: A systematic review and meta-analysis

Ana Brañez-Condorena [1,2], David R. Soriano-Moreno[3], Alba Navarro-Flores[4], Blanca Solis-Chimoy[1,2], Mario E. Diaz-Barrera[5,6], Alvaro Taype-Rondan [7]*

1 Facultad de Medicina, Universidad Nacional Mayor de San Marcos, Lima, Peru, 2 Asociación para el Desarrollo de la Investigación Estudiantil en Ciencias de la Salud (ADIECS), Lima, Peru, 3 Unidad de Investigación Clínica y Epidemiológica, Escuela de Medicina, Universidad Peruana Unión, Lima, Peru, 4 Facultad de Medicina, Universidad Nacional Federico Villarreal, Lima, Peru, 5 Unidad de Investigación en Bibliometría, Universidad San Ignacio de Loyola, Lima, Peru, 6 Sociedad Científica de Estudiantes de Medicina de la Universidad Nacional de Trujillo, SOCEMUNT, Trujillo, Peru, 7 Unidad de Investigación para la Generación y Síntesis de Evidencias en Salud, Universidad San Ignacio de Loyola, Lima, Peru

* alvaro.taype.r@gmail.com

## Abstract

### Background

The Geriatric Depression Scale (GDS) is a widely used instrument to assess depression in older adults. The short GDS versions that have four (GDS-4) and five items (GDS-5) represent alternatives for depression screening in limited-resource settings. However, their accuracy remains uncertain.

### Objective

To assess the accuracy of the GDS-4 and GDS-5 versions for depression screening in older adults.

### Methods

Until May 2020, we systematically searched PubMed, PsycINFO, Scopus, and Google Scholar; for studies that have assessed the sensitivity and specificity of GDS-4 and GDS-5 for depression screening in older adults. We conducted meta-analyses of the sensitivity and specificity of those studies that used the Diagnostic and Statistical Manual of Mental Disorders (DSM) or the International Classification of Diseases-10 (ICD-10) as reference standard. Study quality was assessed with the QUADAS-2 tool. We performed bivariate random-effects meta-analyses to calculate the pooled sensitivity and specificity with their 95% confidence intervals (95% CI) at each reported common cut-off. For the overall meta-analyses, we evaluated each GDS-4 version or GDS-5 version separately by each cut-off, and for investigations of heterogeneity, we assessed altogether across similar GDS

**Data Availability Statement:** All relevant data are within the manuscript and its Supporting Information files.

**Funding:** The author(s) received no specific funding for this work.

**Competing interests:** The authors have declared that no competing interests exist.

versions by each cut-off. Also, we assessed the certainty of evidence using the GRADE methodology.

## Results

Twenty-three studies were included and meta-analyzed, assessing eleven different GDS versions. The number of participants included was 5048. When including all versions together, at a cut-off 2, GDS-4 had a pooled sensitivity of 0.77 (95% CI: 0.70–0.82) and a pooled specificity of 0.75 (0.68–0.81); while GDS-5 had a pooled sensitivity of 0.85 (0.80–0.90) and a pooled specificity of 0.75 (0.69–0.81). We found results for more than one GDS-4 version at cut-off points 1, 2, and 3; and for more than one GDS-5 version at cut-off points 1, 2, 3, and 4. Mostly, significant subgroup differences at different test thresholds across versions were found. The accuracy of the different GDS-4 and GDS-5 versions showed a high heterogeneity. There was high risk of bias in the index test domain. Also, the certainty of the evidence was low or very low for most of the GDS versions.

## Conclusions

We found several GDS-4 and GDS-5 versions that showed great heterogeneity in estimates of sensitivity and specificity, mostly with a low or very low certainty of the evidence. Altogether, our results indicate the need for more well-designed studies that compare different GDS versions.

## Introduction

Depression is a major global public health issue [1]. Older adults represent a vulnerable group, likely due to aging-related factors, such as loss of skills and decreased functional activity [2]. It is estimated that around 10% to 20% of older adults worldwide live with depression [3]. This condition increases the risk of suicide [4], the risk of comorbidities' complications [5], the use of health services and care costs, and overall mortality [4,6]. Hence, it represents a source of high burdening, not only for patients but for healthcare systems.

In older adults, depression´s somatic symptoms are similar to other chronic health conditions [7], and mood changes are less prevalent and commonly replaced by physical discomfort [8,9], resulting in challenging diagnosis and subsequent delay of treatment access. Thus, some structured depression screening scales that focus on elderly population have been developed [10]. There are several scales for screening for depression among older adults, such as the Geriatric Depression Scale (GDS) [11], the Center for Epidemiologic Studies Depression Scale (CES-D), and others. However, the GDS is one of the most used to identify depression among older adults. Among the strengths of the GDS, its use may be easier in people with cognitive impairment because of the simple yes-no format, and it can be used in hospital and community settings [11].

Its full version contains 30 questions (GDS-30) and requires substantial time for assessment. Therefore, shorter GDS versions, selecting some of the GDS-30 items [12,13], have been proposed for a rapid depression assessment in time-restricted scenarios, such as GDS versions with four items (called GDS-4), and GDS versions with five items (called GDS-5) [14–23].

The accuracy of these GDS-4 and GDS-5 versions remains unclear [24]. Although some previous systematic reviews have assessed this subject, these tend to pool different GDS-4 or

GDS-5 versions in the same quantitative analysis, even though each version includes different questions [13,25–27]. Thus, we performed a systematic review that aims to assess the accuracy of the GDS-4 and GDS-5 versions for depression screening in older adults.

## Material and methods

We conducted a systematic review following the Preferred Reporting Items for Systematic Reviews and Meta-Analysis of Diagnostic Test Accuracy Studies (PRISMA-DTA) guidelines [28]. The study protocol is registered in PROSPERO (CRD42020170864).

### Eligibility criteria

The inclusion criteria were as follows: 1) Observational studies that reported the sensitivity and specificity of any of the GDS-4 and GDS-5 versions for the diagnosis of depression, using the DSM or ICD-10 diagnosis criteria as reference standard, since these provide a commonly used and accepted framework for depression diagnosis in the clinical practice [29], 2) studies that were conducted in older adults (at least 2/3 of the study participants must have had ≥ 55 years old), 3) studies that specified the items of the GDS-4 and GDS-5 versions, and 4) studies that provided enough data to construct a 2x2 contingency table to assess sensitivity and specificity. No restrictions on language, publication date, validation of language translation of the short GDS versions, or the mode of test assessment were applied.

### Search strategy

We systematically searched the following databases and search engines: PubMed, PsycINFO and Scopus until April 24, 2020. Additionally, we searched the first 100 results retrieved in Google Scholar up to May 16, 2020. Google Scholar was searched to identify grey literature through the first 100 records, as systematic reviews usually examine the first 100 records in Google Scholar [30–32] because it is a large and unspecific source of grey literature, which sorts results by relevance and coincidence. The search strategy is available at the S1 Table of the Supplementary Material. Later, we complemented the search by reviewing manually the lists of references of all the studies included in the data selection process, the lists of articles that cited each of these included studies (through Google Scholar), and the lists of studies included in previous systematic or narrative reviews on the subject, until May 2020 [13,25–27,33–39].

### Data selection and extraction

Initially, we removed all duplicated records by using the EndNote software. Two independent authors (ANF and DRSM) independently screened all results for inclusion, first reviewing the titles and abstracts, and later performing a full-text assessment, trough EndNote software. Any disagreement during the selection process was discussed with a third party (ABC) and resolved by consensus.

Two authors (ANF and BSC) independently performed the data extraction from each included study using a standardized Microsoft Excel sheet. Differences were solved by a third researcher (ABC). The following variables were extracted: first author, year of publication, country, population characteristics (number of participants, setting, sex, age), inclusion and exclusion criteria, prevalence of depression in the study according to the reference standard, funding, intervention (short GDS version, language of the test, mode of test assessment, GDS-4 or GDS-5 questions, cut-off used, number of true positives, false positives, true negatives, and false negatives), reference standard (International Classification of Diseases [ICD], the

Diagnostic and Statistical Manual of Mental Disorders [DSM], structured interview, or others), type of depression evaluated, and numerical results of sensitivity and specificity. When there were doubts about any information reported in the studies, we sent emails to the authors to clarify the information.

### Risk of bias and certainty of the evidence

Two researchers (DRSM and MEDB) independently assessed the risk of bias of the included studies using the Quality Assessment of Diagnostic Accuracy Studies 2 (QUADAS-2) tool [40]. This tool has four domains: patient selection, index test, reference standard, and flow and timing. The reference standards considered appropriate for this assessment were any version of the DSM or the ICD-10. In case of disagreement, a consensus was achieved with a third researcher (ATR).

Additionally, we used the Grading of Recommendations Assessment, Development, and Evaluation (GRADE) methodology to report the certainty on the evidence [41,42]. Risk of bias, indirect evidence, inconsistency, imprecision, and publication bias were assessed. We downgraded the certainty of evidence when fewer than 70% of studies had at least 7 of 10 items at low risk according to QUADAS-2, when fewer than 70% of studies had the components (population, index test, or reference standard) similar to the initial diagnostic question, when heterogeneity was moderate or high, when the confidence interval range was greater than or equal to 10%, and when fewer than 4 studies evaluated the outcome of interest.

### Statistical analyses

We conducted meta-analyses of the sensitivity and specificity of each of the GDS-4 and GDS-5 versions whenever studies fulfilled the following condition: 1) There was more than one study that compared the same version of GDS-4 or GDS-5 at the same cut-off point. We performed the meta-analyses of GDS-4 and GDS-5 separately.

When there were at least four studies to include in the meta-analysis, we used bivariate mixed-effects models via random effects that consider the correlation between sensitivity and specificity by each study to provide estimates of effects [43]. When less than four studies were included for a meta-analysis, the mixed-effects model assessment was not appropriate, so we performed meta-analyses of proportions using the exact binomial distribution. We calculated the pooled sensitivity and specificity with their 95% confidence intervals.

In addition, we meta-analyzed altogether the results of the included studies that assessed the same cut-off point of any GDS-4 version. Likewise, we meta-analyzed altogether the results of the included studies that assessed the same cut-off point of any GDS-5 version.

Heterogeneity was assessed through visual assessment of forest plots. To assess if there were subgroup differences across different GDS versions, also we evaluated heterogeneity through visual assessment of forest plots. All analyses were performed using the Stata v14.0 software.

### Results

Overall, 2,740 records were retrieved in the database systematic search. After removal of duplicates, 2,254 records were screened, and 71 records were full-text reviewed. From these, we excluded 52 records for not fulfilling the inclusion criteria. Reasons for exclusion are explained in S2 Table. Nineteen records were included in this initial process.

Additionally, we identified seven records that meet our inclusion criteria after searching the lists of references of all included studies, the lists of references of previous reviews, and the lists of articles that cited each of the included studies (through Google Scholar). For a total of 26 included records.

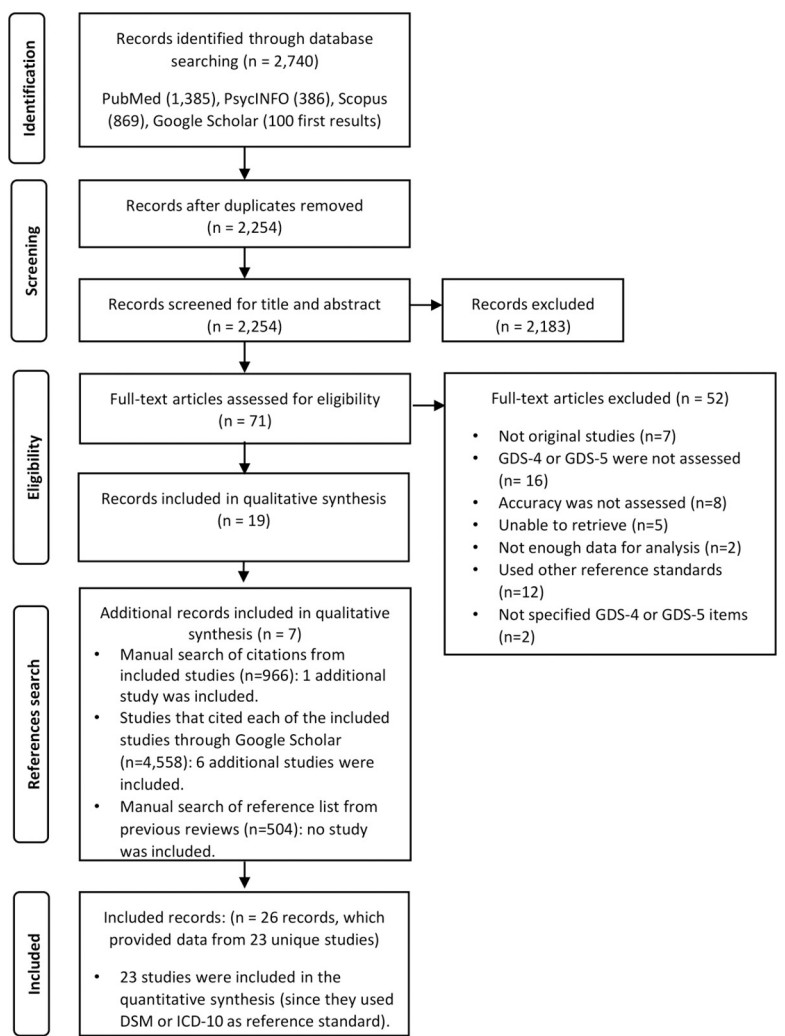

**Fig 1. Flow diagram (study selection).**

Of these, some presented results from the same study: Allgaier 2011 and Allgaier 2013, Castelo 2007 and Castelo 2010, and Cheng 2004 and Cheng 2005 [44–47]. Thus, 26 records representing 23 unique studies were included in the qualitative synthesis. Details of the selection process could be found in Fig 1.

## Studies characteristics

The number of participants included was 5048. Individual studies´ participants ranged from 60 to 586. Regarding the population, one study was performed only in people without dementia [48], and the rest of the studies were performed in both groups of patients [15–19,21–23,44–47,49–61]. Regarding the gold standard used for depression, most studies used the DSM-IV [15,18,22,44–47,49,52,55,56,58,59]. Other standards used were the DSM-III [17,23], DSM-III-R [19,51,60], DSM-IV-TR [54], DSM-V [16,48], and ICD-10 [21,49,53,57,61]. One study did not specify which DSM was evaluated [50]. Nine studies additionally used a structured interview to conduct their assessment such as the Structured Clinical Interview for DSM (SCID) or Composite International Diagnostic Interview (CIDI) [15,23,44–47,49,51,54,56]. The characteristics of the 23 studies are summarized in Table 1 and detailed in S3 Table.

**Table 1. Characteristics of the included studies.**

| Author (Country), Year | Settings | N | Index test | Reference standard |
|---|---|---|---|---|
| Van Marwijk (Netherlands), 1995 [23] | Clinic outpatients | 586 | GDS-4 by Van Marwijk | Major depression and dysthymia assessed by DIS based on DSM-III |
| Almeida (Brazil), 1999 [49] | Clinic outpatients | 64 | GDS-4 by Van Marwijk | Major depressive episode (F32) and dysthymia (F34.1) assessed by ICD-10 checklist of symptoms according to ICD-10, and DSM-IV |
| Hoyl (US), 1999 [15] | Clinic outpatients | 74 | GDS-4 by D'Ath, Van Marwijk GDS-5 by Hoyl | Major depression and depression not otherwise specified assessed by PRIME-MD based on DSM-IV |
| Galaria (US), 2000 [19] | Clinic outpatients | 70 | GDS-4 by Galaria | Major depression assessed by DSM-III-R |
| Chattat (Italy), 2001 [50] | Clinic outpatients | 126 | GDS-5 by Hoyl | Clinical Diagnosis of Depression assessed by DSM (Not specified) |
| De Dios (Spain), 2001 [18] | Clinic outpatients | 155 | GDS-5 by De Dios or Ortega | Major depression, dysthymic disorder, an adaptative disorder with depressive mood and adaptative disorder mixed anxious-depressive assessed by DSM-IV |
| Pomeroy (UK), 2001 [57] | Clinic inpatients | 87 | GDS-4 by D'Ath | A depressive episode (F32) assessed by ICD-10 |
| Rinaldi (Italy), 2003 [58] | Clinic outpatients and nursing home patients | 181 | GDS-5 by Hoyl | Major depression, dysthymia, bipolar depression, and depression not otherwise specified assessed by DSM-IV |
| Cheng (China), 2004 and Cheng, 2005 [17,60] | Clinic outpatients | 444/ 442 | GDS-4 by Cheng | Major depressive disorder, dysthymia, depressive disorder not otherwise specified, adjustment disorder with depressed mood, dementia with depression assessed by DSM-III/Major depression, dysthymia, depressive disorder not otherwise specified, adjustment disorder with depressed mood and dementia with depression assessed by DSM-III-R |
| Jongenelis (Netherlands), 2005 [56] | Nursing homes patients | 333 | GDS-4 by D'Ath, Van Marwijk GDS-5 by Hoyl | Major depression and minor depression assessed by SCAN for DSM-IV |
| Martinez (Spain), 2005 [21] | Clinic outpatients | 249 | GDS-4 by Martinez GDS-5 by Martinez | Clinic diagnosis of depression (Not specified) assessed by ICD-10 |
| De la Torre (Peru), 2006 [52] | Clinic outpatients | 400 | GDS-4 by Galaria | Clinic diagnosis of depression assessed by DSM-IV |
| Castelo (Brazil), 2007 and Castelo, 2010 [46,47] | Clinic outpatients | 220 | GDS-4 by D'Ath, Van Marwijk | Major depressive episode assessed by SCID-I for DSM-IV |
| Izal (Spain), 2007 [54] | Community inhabitants and nursing home patients | 233 | GDS-5 by Hoyl | Major depression assessed by SCID-I for DSM-IV-TR |
| Ortega (Spain), 2007 [22] | Clinic outpatients | 301 | GDS-5 by De Dios or Ortega | Clinical diagnostic of mood disorder assessed by DSM-IV |
| Cheng (China), 2010 [61] | Clinic outpatients | 110 | GDS-4 by Cheng GDS-5 by Cheng or Heisel | Major depression (F32), dysthymia (F34.1), bipolar disorder-depressive episode (F31.3), adjustment disorder with depressed reactions (F43.21), mixed anxiety and depressive disorder (F41.8), dementia with depressive symptoms (F06.31), assessed by ICD-10 |
| Izal (Spain), 2010 [55] | Community inhabitants | 105 | GDS-5 by Hoyl | Major depression assessed by DSM-IV |
| Allgaier (Germany), 2011 and Allgaier, 2013 [44,45] | Nursing homes patients | 92 | GDS-4 by D'Ath | Major depression disorder/Major depression and minor depression assessed by SCID-I for DSM-IV |
| Chin (China), 2014 [51] | Community inhabitants | 388 | GDS-5 by Hoyl (cut-off point was not reported for other versions) | Major depressive disorder assessed by mPDA according to DSM-III-R |
| Apostolo (Portugal), 2018 [16] | Clinic outpatients and inpatients, community inhabitants, and nursing home patients | 139 | GDS-5 by Apostolo | Major depressive episode assessed by DSM-V |
| Dokuzlar (Turkey), 2018 [48] | Community inhabitants | 437 | GDS-4 by Van Marwijk GDS-5 by Hoyl | Major depression assessed by DSM-V |
| Eriksen (Norway), 2019 [53] | Clinic outpatients, community inhabitants, and nursing patients | 194 | GDS-5 by Hoyl | A depressive episode (F32) assessed by ICD-10 |

(*Continued*)

**Table 1.** (Continued)

| Author (Country), Year | Settings | N | Index test | Reference standard |
|---|---|---|---|---|
| Sacuiu (Sweden), 2019 [59] | Clinic outpatients and inpatients, community inhabitants and nursing home patients | 60 | GDS-4 by D'Ath, Van Marwijk<br>GDS-5 by Hoyl, Cheng or Heisel | Major Depressive Disorder assessed by DSM-IV |

DSM: Diagnostic and Statistical Manual of Mental Disorders, ICD: International Classification of Diseases, PRIME-MD: Primary Care Evaluation of Mental Disorders, SCID-I: Structured Clinical Interview for DSM Axis I, DIS: Diagnostic Interview Schedule, SCAN: Schedule of Clinical Assessment in Neuropsychiatry, mPDA: Modified Psychiatrist Diagnostic Assessment.

## GDS-4 and GDS-5 versions

Regarding the short GDS versions, 11 studies only assessed the GDS-4 [17,19,23,44–47,49,52,57,60], 8 studies only assessed the GDS-5 [16,18,22,50,53–55,58], and 7 studies evaluated both of them [15,21,48,51,56,59,61]. We found several GDS-4 and GDS-5 versions that included different items from the original GDS-30. The GDS-4 versions used in the published studies were: D'Ath (n = 7) [15,44,45,47,51,56,57,59], Van Marwijk (n = 8) [15,23,46,48,49,51,56,59], Cheng (n = 2) [17,60,61], Galaria (n = 2) [19,62], Martinez (n = 1) [21] and two authors did not specified which version was used. For the GDS-5, the versions assessed were: Hoyl (n = 9) [15,48,50,51,53–56,58,59], De Dios or Ortega (n = 2) [18,22], Cheng or Heisel (n = 2) [59,61], Molloy (n = 1) [51], Martinez (n = 1) [21], and Apostolo (n = 1) [16].

Each of the GDS-4 and GDS-5 versions assessed different combinations of GDS-30 items. The list of the items assessed by each version is detailed in Table 2. The most assessed questions were the number 1 (satisfied with life) and 3 (life is empty).

## Risk of bias

Using the QUADAS-2 tool, we found a high risk of bias in most of the studies. There was high risk of bias in the index test domain. Specifically, the question about the lack of pre-specification of the cut-off points used was the most common flaw (Fig 2).

## Diagnostic outcomes

As stated before, we assessed the sensitivity and specificity of studies that used the DSM or ICD-10 diagnosis criteria as a reference standard, for all GDS-4 and GDS-5 versions. Thus, 23 studies were included in these quantitative analyses.

**GDS-4.** For the GDS-4 assessment, 14 studies with a total of 3266 participants were included. We obtained eleven sensitivity and specificity estimates, which gave information regarding six versions of GDS-4 at different cut-offs: D'Ath at cut-off 1 and 2; Van Marwijk at cut-off 1, 2 and 3; Cheng at cut-off 1, 2, 3 and 4; Martinez at cut-off 2; and Galaria at cut-off 2 (Table 3).

When taken together, GDS-4 versions at cut-off 1 had a pooled sensitivity of 0.90 (95% CI: 0.85–0.93) and a pooled specificity of 0.57 (95% CI: 0.45–0.67), at cut-off 2 had a pooled sensitivity of 0.77 (95% CI: 0.70–0.82) and a pooled specificity of 0.75 (95% CI: 0.68–0.81), and at cut-off 3 had a pooled sensitivity of 0.63 (95% CI: 0.53–0.71) and a pooled specificity of 0.78 (95% CI: 0.69–0.84).

Among the GDS-4 versions, the results for those with the lower cut-off point tend to have a higher sensitivity and a lower specificity. When assessing the sensitivity and specificity estimates, the Galaria at cut-off 2 and the Cheng at cut-off 4 had the greatest balance, the first one favoring the sensitivity and the second one the specificity.

**Table 2. List of items of each GDS-4 and GDS-5 version found in the included studies.**

| GDS-30 items (over the past week) | GDS-4 | | | | | GDS-5 | | | | | |
|---|---|---|---|---|---|---|---|---|---|---|---|
| | D'Ath | Van Marjwik | Cheng | Galaria | Martinez | Hoyl | De Dios/ Ortega | Cheng/ Heisel | Molloy* | Martinez | Apostolo |
| 1. Satisfied with your life | x | x | | x | | x | x | | x | | x |
| 2. Dropped many of your activities and interests | | x | | x | | | | | | | |
| 3. Your life is empty | x | | | | x | | | x | | x | |
| 4. Often get bored | | | | | x | x | x | | | x | |
| 7. In good spirits most of the time | | | x | | x | | | | | x | x |
| 8. Afraid that something bad is going to happen to you | x | | | | | | | | | | |
| 9. Happy most of the time | x | x | | | | | | x | | x | x |
| 10. Often feel helpless | | | | x | x | x | x | | x | x | |
| 11. Often get restless and fidgety | | | x | | | | | | | | |
| 12. Prefer to stay at home rather than go out and do things | | x | | | | x | x | | | | |
| 14. Have more problems with memory than most | | | | x | | | | | | | |
| 15. Wonderful to be alive now | | x | | | | | | x | | | x |
| 16. Feel downhearted and blue | | x | | | | | | | | | |
| 17. Feel worthless the way you are now | | | | | | x | | x | x | | |
| 20. Hard to get started on new projects | | | | | | | x | | | | |
| 21. Feel full of energy | | | | | | | | | | x | |
| 22. Feel that your situation is hopeless | | | | | | | | x | | | x |

* The assessment of depression is different. If there is a negative answer in item 16, then the evaluation of depression is carried out with the complete GDS-30. If there is a positive answer in item 16, then the evaluation of depression is carried out with the 4 remaining items.

We assessed and found differences in sensitivity and specificity estimates for the different GDS-4 versions, at each cut-off point used.

**GDS-5.** For the GDS-5 assessment, 15 studies with a total of 3085 participants were included. We obtained thirteen sensitivity and specificity estimates, which gave information regarding five versions of GDS-5 at different cut-offs: De Dios or Ortega at cut-off 2, Hoyl at cut-off 1, 2 and 3, Martinez at cut-off 2, Apostolo at cut-off 1, 3, 4 and 5, and Heisel or Cheng at cut-offs 1, 2, 3 and 4 (Table 3).

When taken together, GDS-5 versions at cut-off 1 had a pooled sensitivity of 0.89 (95% CI 0.83–0.94) and a pooled specificity of 0.41 (95% CI 0.30–0.53), at cut-off 2 had a pooled sensitivity of 0.85 (95% CI: 0.80–0.90) and a pooled specificity of 0.75 (95% CI: 0.69–0.81), at cut-off 3 had a pooled sensitivity of 0.60 (95% CI 0.50–0.68) and a pooled specificity of 0.83 (95% CI 0.74–0.89), and at cut-off 4 had a pooled sensitivity of 0.44 (95% CI 0.35–0.53) and a pooled specificity of 0.94 (95% CI 0.88–1.00).

Among the GDS-5 versions, the results for those with the lower cut-off point tend to have a higher sensitivity and a lower specificity. When assessing the sensitivity and specificity estimates, the De Dios or Ortega at cut-off 2 had the greatest balance of sensitivity (0.98, 95% CI: 0.96–1.00) and specificity (0.83, 95% CI: 0.79–0.87).

We assessed and found differences in sensitivity and specificity estimates for the different GDS-5 versions, at each cut-off point used.

A summary of the sensitivity analysis and all the forest plots could be found in S1–S6 Figs.

**Fig 2. Quality assessment using the QUADAS-2 tool.**

## Certainty of evidence

We used GRADE summary of findings (SoF) tables to report the certainty of evidence (Table 3). Overall, the certainty of the evidence was very low, mostly due to concerns about the indirectness of the evidence, inconsistency, and imprecision of the results. However, the De Dios or Ortega GDS-5 version obtained a high certainty of evidence.

## Discussion

The first versions of the GDS-4 and GDS-5 were D'Ath and Hoyl versions, respectively [14,15]. However, many other versions have been created in recent years, mostly by testing which combination of GDS-30 items could have a better performance in terms of sensitivity and specificity [17–20,22,23]. In this systematic review, we found five different versions for the GDS-4 instrument and seven different GDS-5 versions.

Previous systematic reviews have assessed the accuracy of these GDS short versions [13,25–27]. These reviews included from two to ten studies for the GDS-4 assessment, and only one study for the GDS-5 assessment. While in our systematic review we included 23 studies: 15 that evaluated GDS-4 and 15 that evaluated GDS-5.

All previous meta-analysis had pooled the results from studies using different GDS versions. However, results suggest that different versions have different sensitivity and specificity estimates for the same cut-off point.

**Table 3. Summary of diagnostic estimates.**

| Cut-off | GDS version | Studies (n) | Sensitivity (95% CI) | Quality of the Evidence (GRADE) | Specificity (95% CI) | Quality of the Evidence (GRADE) |
|---|---|---|---|---|---|---|
| **GDS-4** | | | | | | |
| 1 | D'Ath | 5 (806) | 0.92 (0.84–0.96) | **VERY LOW**[a,b,c] | 0.61 (0.51–0.70) | **VERY LOW**[b,c,d] |
| | Van Marwijk | 5 (1 650) | 0.92 (0.76–0.97) | **VERY LOW** [d,e,f] | 0.51 (0.28–0.74) | **VERY LOW**[d,e,f] |
| | Cheng | 2 (594) | 0.88 (0.81–0.93) | **LOW**[a,b] | 0.46 (0.38–0.53) | **MODERATE**[b] |
| | **Pooled results** | 9 (2 423) | 0.89 (0.85–0.93) | **MODERATE**[a] | 0.53 (0.42–0.65) | **VERY LOW**[d,f] |
| 2 | D'Ath | 4 (559) | 0.76 (0.61–0.86) | **VERY LOW**[a,e,f,i] | 0.81 (0.69–0.89) | **VERY LOW**[b,d,e,i] |
| | Van Marwijk | 5 (1 275) | 0.79 (0.62–0.90) | **VERY LOW**[c,d,f,j] | 0.72 (0.54–0.85) | **VERY LOW**[c,d,f,j] |
| | Cheng | 2 (594) | 0.73 (0.66–0.80) | **LOW** [a,b] | 0.63 (0.56–0.70) | **MODERATE**[b] |
| | Galaria | 2 (470) | 0.90 (0.84–0.96) | **VERY LOW**[b,c,e] | 0.80 (0.76–0.84) | **LOW**[c,e] |
| | Martinez | 1 (249) | 0.73 (0.64–0.82) | **LOW**[b,h] | 0.78 (0.72–0.84) | **LOW**[b,h] |
| | **Pooled results** | 11(2 740) | 0.77 (0.70–0.82) | **VERY LOW**[a,b,c,e] | 0.75 (0.68–0.81) | **VERY LOW**[b,c,d,e] |
| 3 | Van Marwijk | 1 (64) | 0.85 (0.73–0.97) | **VERY LOW**[f,h] | 0.67 (0.51–0.84) | **VERY LOW**[f,h] |
| | Cheng | 2 (594) | 0.59 (0.53–0.65) | **MODERATE**[b] | 0.79 (0.70–0.85) | **LOW**[a,b] |
| | **Pooled results** | 3 (658) | 0.63 (0.53–0.71) | **LOW**[a,b] | 0.78 (0.69–0.84) | **LOW**[a,b] |
| 4 | Cheng | 1 (444) | 0.79 (0.73–0.85) | **LOW**[b,h] | 0.92 (0.88–0.95) | **MODERATE**[h] |
| **GDS-5** | | | | | | |
| 1 | Cheng or Heisel | 1 (150) | 0.88 (0.82–0.95) | **LOW**[b,h] | 0.38 (0.26–0.51) | **VERY LOW**[f,h] |
| | Apostolo | 1 (139) | 0.91 (0.80–1.00) | **VERY LOW**[b,g,h] | 0.55 (0.46–0.64) | **VERY LOW**[b,g,h] |
| | Hoyl | 1 (333) | 0.93 (0.87–0.99) | **LOW**[b,h] | 0.29 (0.71–0.83) | **LOW**[b,h] |
| | **Pooled results** | 3 (622) | 0.89 (0.83–0.94) | **LOW**[b,c] | 0.41 (0.30–0.53) | **VERY LOW**[c,d,f] |
| 2 | Cheng or Heisel | 1 (150) | 0.78 (0.70–0.86) | **LOW**[b,h] | 0.56 (0.43–0.69) | **VERY LOW**[f,h] |
| | De Dios or Ortega | 2 (456) | 0.98 (0.96–1.00) | **HIGH** | 0.83 (0.79–0.87) | **HIGH** |
| | Hoyl | 9 (2 071) | 0.85 (0.79–0.90) | **VERY LOW**[a,b,e] | 0.77 (0.69–0.83) | **VERY LOW**[b,d,e] |
| | Martinez | 1 (249) | 0.89 (0.73–0.89) | **LOW**[b,h] | 0.73 (0.66–0.80) | **LOW**[b,h] |
| | **Pooled results** | 14 (3 065) | 0.85 (0.80–0.90) | **MODERATE**[a] | 0.75 (0.69–0.81) | **VERY LOW**[b,d] |
| 3 | Apostolo | 1 (139) | 0.78 (0.61–0.95) | **VERY LOW** [f,g,h] | 0.85 (0.79–0.92) | **VERY LOW** [b,g,h] |
| | Cheng or Heisel | 2 (210) | 0.64 (0.54–0.74) | **VERY LOW** [b,c,e] | 0.79 (0.71–0.87) | **VERY LOW** [b,c,e] |
| | Hoyl | 1(333) | 0.77 (0.56–0.97) | **VERY LOW** [e,f,g,h] | 0.86 (0.76–0.96) | **VERY LOW** [b,e,g,h] |
| | **Pooled results** | 3 (622) | 0.60 (0.50–0.68) | **VERY LOW** [b,e,i] | 0.83 (0.74–0.89) | **VERY LOW** [a,b,e,i] |
| 4 | Apostolo | 1 (139) | 0.39 (0.19–0.59) | **VERY LOW**[f,g,h] | 0.97 (0.95–1.00) | **LOW**[g,h] |
| | Cheng or Heisel | 1 (150) | 0.45 (0.35–0.55) | **LOW**[b,h] | 0.90 (0.82–0.97) | **LOW**[b,h] |
| | **Pooled results** | 2 (289) | 0.44 (0.35–0.53) | **LOW**[b,c] | 0.94 (0.88–1.00) | **VERY LOW**[a,b,c] |
| 5 | Apostolo | 1 (139) | 0.13 (0.00–0.27) | **VERY LOW**[f,g,h] | 1.00 (1.00–1.00) | **LOW**[g,h] |

n: Number of participants.

[a] The heterogeneity is moderate.

[b] Wide confidence intervals.

[c] Between 50% and 70% of the studies have similar components to the question.

[d] The heterogeneity is great.

[e] High risk of bias.

[f] Very wide confidence intervals.

[g] The study presents two components similar to the question.

[h] Only one study has been evaluated.

[i] Less than 50% of the studies have similar components (population, index test, or reference standard) to the question.

[j] Very high risk of bias.

Among the assessed GDS-4 versions, the balance between sensitivity and specificity was greater for the Galaria version at cut-off 2 (pooled analysis of two studies, very low certainty of the accuracy evidence), and for the Cheng version at cut-off 4 (one study, low certainty of the accuracy evidence). Among the assessed GDS-5 versions, the balance between sensitivity and specificity was greater for the "De Dios or Ortega" version at cut-off 2 (pooled analysis of two studies with high certainty of the evidence). Although this suggests that the "De Dios or Ortega" version at cut-off 2 may be a balanced option, with a high certainty that allows a more confident estimation of underdiagnosis and overdiagnosis rates, decision-makers must also consider other factors such as applicability in their contexts or cultural variations in the manifestation of depression, before deciding which GDS version to use.

Subgroup analyses found that estimates were different across different GDS-4 versions, and across different GDS-5 versions. While this suggests that some versions may have a better performance than others, the low certainty of these estimates prevents from making any solid conclusion. However, it seems sensible that future systematic reviews evaluate each version separately.

Moreover, most of the meta-analyses for each version also had significant heterogeneity, which may be due to differences in risk of bias, populations characteristics (such as dementia prevalence), study setting, or reference standard usage (DSM-III, DSM-IV, DSM-V, or ICD-10 criteria). Moreover, some cultural differences in the construct of depression may cause heterogeneous results across different contexts [62]. Regretfully, the low number of studies per GDS version and their heterogeneous characteristics prevent to glimpse any predominant factor that could explain the heterogeneous results.

## Limitations and strengths

Certain limitations must be considered when interpreting the results: 1) certainty of the evidence was low or very low for most of the results, mainly due to heterogeneity and risk of bias. 2) Most of studies had a high risk of bias, mainly due to the selective reporting of the cut-off points (some studies seemed to report only the cut-off with the highest sensitivity and specificity), and the assessment of GDS-4 or GDS-5 accuracy by extracting items assessed in a full GDS-30 interview (since the GDS-30 is a much longer survey, it is expected that answering to the GDS-30 would be more exhausted than answering the GDS-4 or GDS-5 versions). 3) Studies had heterogeneous settings, population characteristics, and depression definition.

However, to the best of our knowledge, this is the most comprehensive systematic review performed to date regarding the accuracy of GDS-4 and GDS-5, which included 23 studies; and is the first systematic review that provides the pooled estimates of each GDS-4 and GDS-5 versions. Thus, our results would help guide clinical practice and clinical guidelines recommendations.

## Conclusion

This study summarizes the sensitivity and specificity of GDS-4 and GDS-5 for depression screening in older adults. We found several GDS-4 and GDS-5 versions, the results of which had great heterogeneity, which suggest that some versions may be more accurate than others. Certainty for the evidence was low or very low for almost all estimates. Altogether, our results indicate the need for more well-designed studies that compare different GDS versions.

## Supporting information

**S1 Checklist. PRISMA-DTA checklist item.**
(DOC)

**S1 Fig. D' Ath version.**
(DOCX)

**S2 Fig. Van Marwijk version.**
(DOCX)

**S3 Fig. Cheng version.**
(DOCX)

**S4 Fig. Galaria version.**
(DOCX)

**S5 Fig. Hoyl version.**
(DOCX)

**S6 Fig. Cheng or Heisel version and De Dios or Ortega version.**
(DOCX)

**S1 Table. Search strategy.**
(DOCX)

**S2 Table. Excluded studies.**
(DOCX)

**S3 Table. Characteristics of the included studies.**
(DOCX)

**S1 File.**
(XLSX)

**S1 Database.**
(XLSX)

## Acknowledgments

We would like to thank David Villarreal-Zegarra and Jessica Hanae Zafra-Tanaka for their valuable comments in the revision of the manuscript.

## Author Contributions

**Conceptualization:** Ana Brañez-Condorena, David R. Soriano-Moreno, Blanca Solis-Chimoy, Mario E. Diaz-Barrera, Alvaro Taype-Rondan.

**Data curation:** Ana Brañez-Condorena, David R. Soriano-Moreno, Alba Navarro-Flores, Blanca Solis-Chimoy, Mario E. Diaz-Barrera.

**Formal analysis:** Ana Brañez-Condorena, Alvaro Taype-Rondan.

**Methodology:** Ana Brañez-Condorena, David R. Soriano-Moreno, Alba Navarro-Flores, Blanca Solis-Chimoy, Mario E. Diaz-Barrera, Alvaro Taype-Rondan.

**Supervision:** Alvaro Taype-Rondan.

**Visualization:** Alba Navarro-Flores, Alvaro Taype-Rondan.

**Writing – original draft:** Ana Brañez-Condorena, David R. Soriano-Moreno, Alba Navarro-Flores, Blanca Solis-Chimoy, Mario E. Diaz-Barrera, Alvaro Taype-Rondan.

**Writing – review & editing:** Ana Brañez-Condorena, David R. Soriano-Moreno, Alba Navarro-Flores, Blanca Solis-Chimoy, Mario E. Diaz-Barrera, Alvaro Taype-Rondan.

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
