## [Decision Letter · Decision Letter 0]

22 Mar 2021

PONE-D-21-03547

Accuracy of the Geriatric Depression Scale (GDS)-4 and GDS-5 for the screening of depression among older adults: a systematic review and meta-analysis

PLOS ONE

Dear Alvaro Taype-Rondan, 

Thank you for submitting your manuscript to PLOS ONE. After careful consideration, we feel that it has merit but does not fully meet PLOS ONE’s publication criteria as it currently stands. Therefore, we invite you to submit a revised version of the manuscript that addresses the points raised during the review process.

This is an interesting review but the rationale and methodology(especially) warrant considerable revisions. Please refer to references provided in the peer-review feedback on resources that can guide the correct methods for conducting and reporting systematic reviews and meta-analyses of Diagnostic Accuracy studies. One example is the PRISMA statement for diagnostic accuracy reviews which was not used and referenced by the author team.

We look forward to receiving your revised manuscript.

Kind regards,

Eleanor Ochodo, M.D., PhD

Academic Editor

PLOS ONE

Journal Requirements:

2. Please include a copy of Table 4 which you refer to in your text on page 13.

3. Please include captions for *all* Supporting Information files at the end of your manuscript, and update any in-text citations to match accordingly. Please see our Supporting Information guidelines for more information: http://journals.plos.org/plosone/s/supporting-information.

Reviewers' comments:

Reviewer's Responses to Questions

**Comments to the Author**

1. Is the manuscript technically sound, and do the data support the conclusions?

Reviewer #1: Yes

Reviewer #2: Partly

2. Has the statistical analysis been performed appropriately and rigorously? 

Reviewer #1: Yes

Reviewer #2: No

3. Have the authors made all data underlying the findings in their manuscript fully available?

Reviewer #1: Yes

Reviewer #2: Yes

4. Is the manuscript presented in an intelligible fashion and written in standard English?

Reviewer #1: Yes

Reviewer #2: Yes

5. Review Comments to the Author

Reviewer #1: Overall, the authors have done rigorous work to examine the utility of the brief versions of GDS (GDS-5 and 4) in screening for geriatric depression. Despite the challenge of inconsistency in the item used and the cut-off value, they managed to segregate the various versions of GDS4 and 5 with the same cut-off point and performed meta-analysis whenever possible. The outcome is summarized as pooled sensitivity and specificity with the Youden index. Since2*2 contingency table is constructed it would be more informative if the authors could report other measures of diagnostic accuracy like a pooled estimate of the positive and negative likelihood ratios, summary diagnostic odds ratio and area under curve (AOC).

Major:

Methods: For the outcome measures, since 2 by 2 contingency table is constructed, it would be better if authors could include other measures of diagnostic accuracy like a pooled estimate of the positive and negative likelihood ratios, summary diagnostic odds ratio, and area under curve (AOC).

Furthermore, I wonder if Cronbach’s alfa of the items enlisted in table 2 could be summarized to recommend for future studies.

Minor:

Line 60: it is combining? Or just selecting some items from the original 30-item version?

Line 62,63: word different and altogether may not be needed.

Line 147: Regarding the population, …. The sentence needs reconstruction. Only one study excluded the sample with dementia.

Line 74-81: Authors have elaborated the various eligibility criteria for inclusion. Authors can further clarify:

1.      Language of the tool. If translated, whether the translation was done appropriately

2.      I understood that irrespective of Study design, setting, the way the interview was included. It may be better to mention.

Reviewer #2: This review aims to assess the diagnostic accuracy of Geriatric Depression Scale (GDS)-4 and GDS-5 for screening of depression among older adults

General comments

This is an interesting paper but the rationale needs to be strengthened and methods revised considerably. The methodology is presents some inaccuracies and confusion with methods used for meta-analyses of intervention reviews. I would recommend the following references to the authors;

• PRISMA for Diagnostic Test Accuracy Reviews (DTA): http://www.prisma-statement.org/Extensions/DTA

• Leeflang MM. Systematic reviews and meta-analyses of diagnostic test accuracy. Clin Microbiol Infect. 2014 Feb;20(2):105-13. doi: 10.1111/1469-0691.12474. PMID: 24274632 https://pubmed.ncbi.nlm.nih.gov/24274632/

• Leeflang MM, Deeks JJ, Gatsonis C, Bossuyt PM; Cochrane Diagnostic Test Accuracy Working Group. Systematic reviews of diagnostic test accuracy. Ann Intern Med. 2008 Dec 16;149(12):889-97. doi: 10.7326/0003-4819-149-12-200812160-00008. https://pubmed.ncbi.nlm.nih.gov/19075208/

• DTA meta-analyses methods: Chapter 10 (Analysis of results) of the Cochrane handbook for DTA reviews. https://methods.cochrane.org/sdt/handbook-dta-reviews

• GRADE: https://gradepro.org/ GRADE Pro /GDT is a free software that enables authors conduct GRADE assessments accurately and also generates nice summary of findings tables. (Table 3 needs to follow this format)

Specific comments

ABSTRACT

Methods section

• This statement in the methods section is unclear. “We conducted sensitivity and specificity meta-analyses of those studies using the Diagnostic and Statistical Manual of Mental Disorders (DSM) or the International Classification of Diseases-10 (ICD-10). Do the authors mean using “Diagnostic and Statistical Manual of Mental Disorders (DSM) or the International Classification of Diseases-10 (ICD-10)” as the reference standard? In its current form, the statement implies that these were the statistical methods used to pool the estimates of sensitivity and specificity.

• “Mostly, significant subgroup differences across versions were found”. It would be helpful to state which subgroups were measured. Do the authors means subgroup differences at different test thresholds?

Conclusion

• This statement in the conclusion is not qualified in the results section. “Conclusions: The accuracy of the different GDS-4 and GDS-5 versions showed a high heterogeneity”. This statement is best placed in the results section.

MAIN TEXT

Introduction

• The rationale, “accuracy remains unclear” is a vague rationale. Is the lack of clarity due to variation in accuracy estimates of existing primary studies? Please qualify that rationale better. I also disagree that the existing published systematic reviews are all outdated. Some were published in 2017 (ref 27) and 2019 (ref 26). These are recent reviews. An outdated review is usually > 5years.

• It would be good to include a paragraph in the rationale about the best available test/reference test- what it is, its strengths and limitations as well as the anticipated role of the index tests. Are the GDS scales being evaluated as replacement tests for the reference tests/existing tests?

Methods

• The reporting of this review would be greatly improved if the authors were guided by the PRISMA extension for Diagnostic Test Accuracy reviews and not the original PRISMA. Please re-write this review based on PRISMA DTA.[ McInnes MDF, Moher D, Thombs BD, McGrath TA, Bossuyt PM; and the PRISMA-DTA Group; Clifford T, Cohen JF, Deeks JJ, Gatsonis C, Hooft L, Hunt HA, Hyde CJ, Korevaar DA, Leeflang MMG, Macaskill P, Reitsma JB, Rodin R, Rutjes AWS, Salameh JP, Stevens A, Takwoingi Y, Tonelli M, Weeks L, Whiting P, Willis BH. Preferred Reporting Items for a Systematic Review and Meta-analysis of Diagnostic Test Accuracy Studies: The PRISMA-DTA Statement. JAMA. 2018;319(4):388-396.]

Search strategy

• Please clarify why only the first 100 results yielded in google scholar were searched. Google scholar generates lots of hits. How did the authors ensure that the first 100 were the most relevant to screen?

Data selection and extraction.

• The authors state that duplicates were removed using endnote reference tool and data extraction was done using an excel sheet. Please clarify which platform/software specifically was used to screen titles, abstracts and full texts of the search yield?

Risk of bias and certainly of evidence

• Please add more detail about how GRADE was used to assess certainty of evidence? How was downgrading done?

Statistical analyses

• This section needs to be revised for clarity. Please refer to Chapter 10 (Analysis section) in the Cochrane handbook for DTA reviews. https://methods.cochrane.org/sdt/handbook-dta-reviews

• Please provide a reference to qualify the type of meta-analyses used (bivariate model).Also clarify if it was the bivariate random effects method (which is commonly used) or bivariate mixed-effects models. By mixed effects do you mean random and fixed effects combined?

• Please state clearly at the beginning of this section that meta-analyses of GDS-4 and GDS-5 were done separately

• Please provide a rationale why the Y index was calculated provided. This is a global measure of accuracy and to my knowledge rarely used nowadays because of its limitations.

• Please revisit how heterogeneity is measured in DTA reviews. I2 is used to assess heterogeneity of intervention reviews and not recommended for DTA reviews.

Results

• The results section about risk of bias is thin. QUADAS has four domains against which risk of bias results are reported. Please specify which domains were deemed to have risk of bias.

• Table 3. The reporting of GRADE results is incorrect. GRADE assessment is given for an overall summary of evidence and not individual studies as presented. QUADAS is for individual studies but GRADE summarises the overall certainly of evidence across the domains quality/risk of bias; inconsistency, imprecision, indirectness and publication bias. For example, one would except an overall certainly of evidence for pooled results at each cutoff but not for individual studies. Please refer to the GRADE pro software to help with the GRADE assessment as well as generation of an accurate summary of findings table (https://gradepro.org/).

6. PLOS authors have the option to publish the peer review history of their article (what does this mean?). If published, this will include your full peer review and any attached files.

Reviewer #1: **Yes: **Roshana Shrestha

Reviewer #2: No

---

## [Author Response · Author response to Decision Letter 0]

23 Apr 2021

Dear editor and reviewers,

Thank you for your kind consideration. In this letter, we proceed to respond to each of the comments made by the reviewers and the journal requirements.

Sincerely, 

Alvaro Taype-Rondan, corresponding author

Journal Requirements:

Answer. Thank you for your observation. We have checked the requirements and our manuscript meets them.

2. Please include a copy of Table 4 which you refer to in your text on page 13.

Answer. Thank you for your observation. In our manuscript, there are 3 tables in total. "Table 4" should not appear. On page 13, we have replaced “Table 4” with “Table 3”.

3. Please include captions for *all* Supporting Information files at the end of your manuscript, and update any in-text citations to match accordingly. Please see our Supporting Information guidelines for more information: http://journals.plos.org/plosone/s/supporting-information.

Answer. Thank you for your observation. We added the captions for Supporting Information files at the end of the manuscript. Also, we have updated the in-text citations.

Reviewers' comments:

R1.1: Methods: For the outcome measures, since 2 by 2 contingency table is constructed, it would be better if authors could include other measures of diagnostic accuracy like a pooled estimate of the positive and negative likelihood ratios, summary diagnostic odds ratio, and area under curve (AOC).

Answer. Thank you for your recommendation. For decision making in clinical practice, sensitivity and specificity are used to identify false positives, false negatives, true positives, and true negatives, and thus to consider the desirable and undesirable consequences of diagnostic tests for patients (1). Although we could include the other diagnostic measures, we believe that it would enlarge the table, and make it less understandable. Thus, we chose to keep sensitivity and specificity as the most useful and sufficient indicators for decision-making.

1. Buehler AM, Ascef BO, Oliveira Júnior HA, Ferri CP, Fernandes JG. Rational use of diagnostic tests for clinical decision making. Rev Assoc Med Bras. 2019;65(3):452-459. doi: 10.1590/1806-9282.65.3.452. 

R1.2: Furthermore, I wonder if Cronbach’s alfa of the items enlisted in table 2 could be summarized to recommend for future studies.

Answer. Thank you for your observation. The included studies provided data to construct the 2x2 table for each short GDS version but did not provide the variance data of each short GDS version to calculate Cronbach's alpha (1). The sensitivity and specificity data did not allow us to calculate the correlation between items, hence we were unable to compute Cronbach's alpha.

1. Cortina J.What is coefficient alpha: an examination of theory and applications. J Appl Psychol. 1993;78:98-104. doi: 10.1037/0021-9010.78.1.98

R1.3: Line 60: it is combining? Or just selecting some items from the original 30-item version?

Answer. Thank you for your observation. We have corrected the redaction of that sentence. We have changed “Therefore, shorter GDS versions, combining some of the GDS-30 items” to “Therefore, shorter GDS versions, selecting some of the GDS-30 items”.

R1.4: Line 62,63: word different and altogether may not be needed.

Answer. Thank you for your observation. We have changed “such as different GDS versions with four items (called altogether GDS-4), and different GDS versions with five items (called altogether GDS-5)” to “such as GDS versions with four items (called GDS-4), and GDS versions with five items (called GDS-5)”.

R1.5: Line 147: Regarding the population, …. The sentence needs reconstruction. Only one study excluded the sample with dementia.

Answer. Thank you for your observation. We have rewritten that sentence now in Line 166: “Regarding the population, one study was performed only in people without dementia [44], and the rest of the studies were performed in both groups of patients”.

R1.6: Line 74-81: Authors have elaborated the various eligibility criteria for inclusion. Authors can further clarify:

1. Language of the tool. If translated, whether the translation was done appropriately

2. I understood that irrespective of Study design, setting, the way the interview was included. It may be better to mention.

Answer. Thank you for your observation. The GDS-4 and GDS-5 versions of the included studies were translated into different languages and applied. Although it is assumed that these instruments were validated prior to their use for depression screening, there were studies that explicitly stated that they used the validated instrument, while others did not state their validation information. We included studies that evaluated the GDS-4 or GDS-5 versions independent of whether they detailed validation in the language of the study participants.

Regarding the study design, we have included observational studies as those are adequately designed for the assessment of diagnostic tests. Also, we had no restrictions of setting and interview. We added the following lines in “Eligibility criteria” section Line 87-89: No restrictions on language, publication date, validation of language translation of the short GDS versions, or the mode of test assessment were applied”.

In S3 Table, we added the following data: “Language of the test” and “Mode of test assessment”.

R2.1: This is an interesting paper but the rationale needs to be strengthened and methods revised considerably. The methodology is presents some inaccuracies and confusion with methods used for meta-analyses of intervention reviews. I would recommend the following references to the authors;

• PRISMA for Diagnostic Test Accuracy Reviews (DTA): http://www.prisma-statement.org/Extensions/DTA

• Leeflang MM. Systematic reviews and meta-analyses of diagnostic test accuracy. Clin Microbiol Infect. 2014 Feb;20(2):105-13. doi: 10.1111/1469-0691.12474. PMID: 24274632 https://pubmed.ncbi.nlm.nih.gov/24274632/

• Leeflang MM, Deeks JJ, Gatsonis C, Bossuyt PM; Cochrane Diagnostic Test Accuracy Working Group. Systematic reviews of diagnostic test accuracy. Ann Intern Med. 2008 Dec 16;149(12):889-97. doi: 10.7326/0003-4819-149-12-200812160-00008. https://pubmed.ncbi.nlm.nih.gov/19075208/

• DTA meta-analyses methods: Chapter 10 (Analysis of results) of the Cochrane handbook for DTA reviews. https://methods.cochrane.org/sdt/handbook-dta-reviews

• GRADE: https://gradepro.org/ GRADE Pro /GDT is a free software that enables authors conduct GRADE assessments accurately and also generates nice summary of findings tables. (Table 3 needs to follow this format)

Answer. Thank you for your observation. We have read the suggested literature on systematic reviews and meta-analyses of diagnostic test accuracy to improve the methodology in our manuscript. We have added the following lines in the Abstract as required by PRISMA-DTA: "Study quality was assessed with the QUADAS-2 tool”, “We performed a bivariate random-effects meta-analysis to calculate the pooled sensitivity and specificity with their 95% confidence intervals (95% CI)", "The number of participants included was 5048", and "There was high risk of bias in the index test domain". In addition, with the corrections made in the other reviewers' comments, we comply with the rest of the content indicated in the PRISMA-DTA. In the "Material and Methods" section, we have changed "We conducted a systematic review following the Preferred Reporting Items for Systematic Reviews and Meta-Analyses (PRISMA) guidelines" to "We conducted a systematic review following the Preferred Reporting Items for Systematic Reviews and Meta-Analyses of Diagnostic Test Accuracy Studies (PRISMA-DTA) guidelines [28]".

28. McInnes MDF, Moher D, Thombs BD, McGrath TA, Bossuyt PM, Clifford T, et al. Preferred Reporting Items for a Systematic Review and Meta-analysis of Diagnostic Test Accuracy Studies: The PRISMA-DTA Statement. JAMA. 2018;319:388-396. doi: 10.1001/jama.2017.19163.

Regarding Table 3, the Summary of Findings table format for GRADE Pro diagnostic questions does not seem to us to be the most convenient because the data on the prevalence of depression in the elderly can be very variable among different subsets of elderly people (1,2). Therefore, we opted to continue with the format of Table 3 where we place the certainty of evidence evaluated according to the GRADE methodology.

1. Djernes JK. Prevalence and predictors of depression in populations of elderly: a review. Acta Psychiatr Scand. 2006;113(5):372-87. doi: 10.1111/j.1600-0447.2006.00770.x. PMID: 16603029.

2. Blazer DG. Depression in late life: review and commentary. J Gerontol A Biol Sci Med Sci. 2003;58(3):249-65. doi: 10.1093/gerona/58.3.m249. PMID: 12634292.

R2.2: ABSTRACT

Methods section

• This statement in the methods section is unclear. “We conducted sensitivity and specificity meta-analyses of those studies using the Diagnostic and Statistical Manual of Mental Disorders (DSM) or the International Classification of Diseases-10 (ICD-10). Do the authors mean using “Diagnostic and Statistical Manual of Mental Disorders (DSM) or the International Classification of Diseases-10 (ICD-10)” as the reference standard? In its current form, the statement implies that these were the statistical methods used to pool the estimates of sensitivity and specificity.

Answer. Thank you for your observation. We meant that DSM and ICD-10 were used as reference standard. In “Methods” section of the abstract, we have changed “We conducted sensitivity and specificity meta-analyses of those studies using the Diagnostic and Statistical Manual of Mental Disorders (DSM) or the International Classification of Diseases-10 (ICD-10)” to “We conducted sensitivity and specificity meta-analyses of those studies that used the Diagnostic and Statistical Manual of Mental Disorders (DSM) or the International Classification of Diseases-10 (ICD-10) as reference standard”. 

R2.3: “Mostly, significant subgroup differences across versions were found”. It would be helpful to state which subgroups were measured. Do the authors means subgroup differences at different test thresholds?

Answer. Thank you for your observation. We had specified which subgroups were analyzed, each subgroup is defined by a different GDS-4/-5 version and a threshold. We have added that explanation in the “Results” section of the abstract: “Mostly, significant subgroup differences at different test thresholds across versions were found”.

R2.4: Conclusion

• This statement in the conclusion is not qualified in the results section. “Conclusions: The accuracy of the different GDS-4 and GDS-5 versions showed a high heterogeneity”. This statement is best placed in the results section.

Answer. Thank you for your observation. We added the following line in “Results” section of the abstract: “The accuracy of the different GDS-4 and GDS-5 versions showed a high heterogeneity”. In “Conclusions” section of the abstract, we added the following lines: “We found several GDS-4 and GDS-5 versions that showed great heterogeneity, mostly with a low or very low certainty of the evidence”.

R2.5: MAIN TEXT

Introduction

• The rationale, “accuracy remains unclear” is a vague rationale. Is the lack of clarity due to variation in accuracy estimates of existing primary studies? Please qualify that rationale better. I also disagree that the existing published systematic reviews are all outdated. Some were published in 2017 (ref 27) and 2019 (ref 26). These are recent reviews. An outdated review is usually > 5years.

Answer. Thank you for your observation. We have developed the idea of our rationale as follows: Previous systematic reviews had wrongly focused their analysis, by pooling the results of different versions of each scale (GDS-4/-5) that represented different instruments as the used different questions from the GDS-30, and evaluated together the diagnostic values obtained by different scales. This analysis was incorrect as there are different types of versions of GDS-4 and GDS-5 and they should not be evaluated as one. Thus, although there are previous recent systematic reviews with metanalyses [26,27], the actual accuracy of the GDS-4/GDS-5 scales remains unclear. 

In “Introduction” section, we have changed “Although some previous systematic reviews have assessed this subject, these are outdated and tend to pool…” to “Although some previous systematic reviews have assessed this subject, these tend to pool…”

26. Krishnamoorthy Y, Rajaa S, Rehman T. Diagnostic accuracy of various forms of geriatric depression scale for screening of depression among older adults: Systematic review and meta-analysis. Archives of Gerontology and Geriatrics. 2020;87:104002. doi: https://doi.org/10.1016/j.archger.2019.104002.

27. Tsoi KK, Chan JY, Hirai HW, Wong SY. Comparison of diagnostic performance of Two-Question Screen and 15 depression screening instruments for older adults: systematic review and meta-analysis. The British journal of psychiatry : the journal of mental science. 2017;210(4):255-60. Epub 2017/02/18. doi: 10.1192/bjp.bp.116.186932. PubMed PMID: 28209592.

R2.6: It would be good to include a paragraph in the rationale about the best available test/reference test- what it is, its strengths and limitations as well as the anticipated role of the index tests. Are the GDS scales being evaluated as replacement tests for the reference tests/existing tests?

Answer. Thank you for your observation. We added the following lines in “Introduction” section: “There are several scales for screening for depression among older adults, such as the Geriatric Depression Scale (GDS) [11], the Center for Epidemiologic Studies Depression Scale (CES-D), and others. However, the GDS is one of the most used to identify depression among older adults [11]. Among the strengths of the GDS, its use may be easier in people with cognitive impairment because of the simple yes-no format, and it can be used in hospital and community settings”.

[11] O'Connor E, Rossom RC, Henninger M, Groom HC, Burda BU, Henderson JT, et al. U.S. Preventive Services Task Force Evidence Syntheses, formerly Systematic Evidence Reviews. Screening for Depression in Adults: An Updated Systematic Evidence Review for the US Preventive Services Task Force. Rockville (MD): Agency for Healthcare Research and Quality (US); 2016.

R2.7: Methods

• The reporting of this review would be greatly improved if the authors were guided by the PRISMA extension for Diagnostic Test Accuracy reviews and not the original PRISMA. Please re-write this review based on PRISMA DTA.[ McInnes MDF, Moher D, Thombs BD, McGrath TA, Bossuyt PM; and the PRISMA-DTA Group; Clifford T, Cohen JF, Deeks JJ, Gatsonis C, Hooft L, Hunt HA, Hyde CJ, Korevaar DA, Leeflang MMG, Macaskill P, Reitsma JB, Rodin R, Rutjes AWS, Salameh JP, Stevens A, Takwoingi Y, Tonelli M, Weeks L, Whiting P, Willis BH. Preferred Reporting Items for a Systematic Review and Meta-analysis of Diagnostic Test Accuracy Studies: The PRISMA-DTA Statement. JAMA. 2018;319(4):388-396.]

Answer. Thank you for your observation. We have added the following lines in the Abstract as required by PRISMA-DTA: "Study quality was assessed with the QUADAS-2 tool”, “We performed a bivariate random-effects meta-analysis to calculate the pooled sensitivity and specificity with their 95% confidence intervals (95% CI)", "The number of participants included was 5048", and "There was high risk of bias in the index test domain". In addition, with the corrections made in the other reviewers' comments, we comply with the rest of the content indicated in the PRISMA-DTA. In the "Material and Methods" section, we have changed "We conducted a systematic review following the Preferred Reporting Items for Systematic Reviews and Meta-Analyses (PRISMA) guidelines" to "We conducted a systematic review following the Preferred Reporting Items for Systematic Reviews and Meta-Analyses of Diagnostic Test Accuracy Studies (PRISMA-DTA) guidelines [28]".

28. McInnes MDF, Moher D, Thombs BD, McGrath TA, Bossuyt PM, Clifford T, et al. Preferred Reporting Items for a Systematic Review and Meta-analysis of Diagnostic Test Accuracy Studies: The PRISMA-DTA Statement. JAMA. 2018;319:388-396. doi: 10.1001/jama.2017.19163.

R2.8: Search strategy

• Please clarify why only the first 100 results yielded in google scholar were searched. Google scholar generates lots of hits. How did the authors ensure that the first 100 were the most relevant to screen?

Answer. Thank you for your observation. As Google Scholar is a big and unspecific source of grey literature, which orders the results by relevance and coincidence, Systematic reviews usually examine the first results (usually the first 100 records) in Google Scholar (1-3). We searched Google Scholar to identify grey literature through the first 100 records. We entered the Internet as incognito and removed the cache (data previously stored on the computer) so that previous data would have no influence on the order of appearance of the search results. These results were ordered according to relevance and were not restricted by publication date. The first 100 records were evaluated. In “Search strategy” section, we added the following lines (Line 93-100): “Google Scholar was searched to identify grey literature through the first 100 records, as systematic reviews usually examine the first 100 records in Google Scholar because it is a large and unspecific source of grey literature, which sorts results by relevance and coincidence”.

1. Abdullahi A, Candan SA, Abba MA, Bello AH, Alshehri MA, Afamefuna Victor E, et al. Neurological and Musculoskeletal Features of COVID-19: A Systematic Review and Meta-Analysis. Front Neurol. 2020;11:687. doi: 10.3389/fneur.2020.00687

2. Joyeux L, De Bie F, Danzer E, Russo FM, Javaux A, Peralta CFA, et al. Learning curves of open and endoscopic fetal spina bifida closure: systematic review and meta-analysis. Ultrasound Obstet Gynecol. 2020;55(6):730–9. doi: 10.1002/uog.20389

3. Arora M, Chugh A, Jain N, Mishu M, Boeckmann M, Dahanayake S, et al. Global impact of tobacco control policies on smokeless tobacco use: a systematic review protocol. BMJ Open. 2020 Dec 24;10(12):e042860.

doi: 10.1136/bmjopen-2020-042860

R2.9: Data selection and extraction.

• The authors state that duplicates were removed using endnote reference tool and data extraction was done using an excel sheet. Please clarify which platform/software specifically was used to screen titles, abstracts and full texts of the search yield?

Answer. Thank you for your observation. Screening of titles, abstracts, and full texts was performed manually in Endnote. After eliminating duplicates, two Endnote libraries were created with the records that were found so that two people could independently perform the screening, and compare their selection of studies. We added the following lines in “Data selection and extraction” section (lines 102-104): “Two independent authors (ANF and DRSM) independently screened all results for inclusion, first reviewing the titles and abstracts, and later performing a full-text assessment, through EndNote software”.

R2.10: Risk of bias and certainly of evidence

• Please add more detail about how GRADE was used to assess certainty of evidence? How was downgrading done?

Answer. Thank you for your observation. We added the following lines in “Risk of bias and certainty of evidence” section (Lines 125-131): “Risk of bias, indirect evidence, inconsistency, imprecision, and publication bias were assessed. We downgraded the certainty of evidence when fewer than 70% of studies had at least 7 of 10 items at low risk according to QUADAS-2, when fewer than 70% of studies had the components (population, index test, or reference standard) similar to the initial diagnostic question, when heterogeneity was moderate or high, when the confidence interval range was greater than or equal to 10%, and when fewer than 4 studies evaluated the outcome of interest”.

R2.11: Statistical analyses

• This section needs to be revised for clarity. Please refer to Chapter 10 (Analysis section) in the Cochrane handbook for DTA reviews. https://methods.cochrane.org/sdt/handbook-dta-reviews

Answer. Thank you for your observation. The process of data analysis was performed according to the Cochrane handbook (allowing analysis replication), as follows:

First, we installed the package “ssc install midas” and “ssc install metaprop” in STATA, which were the packages of diagnostic and proportion meta-analysis, respectively. Then, we input the commands “midas tp fp fn tn id(author year) ms(0.75) ford fors bfor(dss)” and got the forest plots for sensitivity and specificity. When less than four studies were included for a meta-analysis, we performed meta-analyses of proportions. We input the commands “metaprop num denom, random” and got the meta-analysis of proportions. 

In “Statistical analyses” section, we added the following lines (Lines 137-147): “When there were at least four studies to include in the meta-analysis, we used bivariate mixed-effects models via random effects that consider the correlation between sensitivity and specificity by each study to provide estimates of effects [40]. When less than four studies were included for a meta-analysis, the mixed-effects model assessment was not appropriate, so we performed meta-analyses of proportions using the exact binomial distribution. We calculated the pooled sensitivity and specificity with their 95% confidence intervals” and “Heterogeneity was assessed through visual assessment of forest plots”

R2.12: Please provide a reference to qualify the type of meta-analyses used (bivariate model). Also clarify if it was the bivariate random effects method (which is commonly used) or bivariate mixed-effects models. By mixed effects do you mean random and fixed effects combined?

Answer. Thank you for your observation. We used a “bivariate mixed-effects regression framework focused on making inferences about average sensitivity and specificity” through midas command in Stata for meta-analysis for diagnostic test performance (1). This mixed model is part of the bivariate random effects method (2). We added the following line in “Statistical analyses” section (Lines 137-139): “When there were at least four studies to include in the meta-analysis, we used bivariate mixed-effects models via random effects that consider the correlation between sensitivity and specificity by each study to provide estimates of effects [43]”.

1. Dwamena BA, Sylvester R, Carlos RC. MIDAS: Stata module for meta-analytical integration of diagnostic test accuracy studies. Statistical Software Components [Internet]. 2009 [cited 2021 Apr 16] Available from: http://fmwww.bc.edu/repec/bocode/m/midas.pdf

2. Reitsma JB, Glas AS, Rutjes AW, Scholten RJ, Bossuyt PM, Zwinderman AH. Bivariate analysis of sensitivity and specificity produces informative summary measures in diagnostic reviews. J Clin Epidemiol. 2005;58(10):982-90. doi: 10.1016/j.jclinepi.2005.02.022. PMID: 16168343.

R2.13: Please state clearly at the beginning of this section that meta-analyses of GDS-4 and GDS-5 were done separately

Answer. Thank you for your observation. We added the following line in “Statistical analyses” section: “We performed the meta-analyses of GDS-4 and GDS-5 separately”.

R2.14: Please provide a rationale why the Y index was calculated provided. This is a global measure of accuracy and to my knowledge rarely used nowadays because of its limitations.

Answer. Thank you for your observation. As you rightly pointed out, the Youden index is a measure currently no longer used since it has several limitations. We found that adding the Youden index to the manuscript did not add anything new and the results were still understood despite of it. We therefore chose to delete the Youden index from the manuscript. We have removed all content about Youden Index in the "Statistical analysis", "Results", "Discussion", and "Table 3" sections.

R2.15: Please revisit how heterogeneity is measured in DTA reviews. I2 is used to assess heterogeneity of intervention reviews and not recommended for DTA reviews.

Answer. Thank you for your observation. We agree with you. The Cochrane Handbook for DTA reviews (1) indicates that it is not appropriate to use the i2 for sensitivity and specificity as it would overestimate the degree of heterogeneity observed. It also indicates that heterogeneity is usually assessed through visual assessment of forest plots and in ROC space. They indicate that heterogeneity can be observed through estimates with confidence intervals. When the results of the studies are different, there is little overlap in the confidence intervals. With respect to the ROC space, they indicate that it is rarely possible to obtain confidence intervals. It is impossible to assess whether the differences between studies are within the expected limit due to chance or due to real differences between studies. Therefore, we opted for the evaluation of heterogeneity through visual assessment of forest plots. We have removed the heterogeneity assessment with i2 in the manuscript and assessed heterogeneity visually with forest plots. We changed all content about I2 in the "Statistical analysis", "Results", and "Table 3" sections.

1. DTA meta-analyses methods: Chapter 11 (Interpreting results and drawing conclusions) of the Cochrane handbook for DTA reviews. https://methods.cochrane.org/sdt/handbook-dta-reviews

R2.16: Results

• The results section about risk of bias is thin. QUADAS has four domains against which risk of bias results are reported. Please specify which domains were deemed to have risk of bias.

Answer. Thank you for your observation. We added the following lines in “Results” section (Lines 199-201): “There was high risk of bias in the index test domain. Specifically, the question about the lack of pre-specification of the cut-off points was the most common flaw”.

R2.17: Table 3. The reporting of GRADE results is incorrect. GRADE assessment is given for an overall summary of evidence and not individual studies as presented. QUADAS is for individual studies but GRADE summarises the overall certainly of evidence across the domains quality/risk of bias; inconsistency, imprecision, indirectness and publication bias. For example, one would except an overall certainly of evidence for pooled results at each cutoff but not for individual studies. Please refer to the GRADE pro software to help with the GRADE assessment as well as generation of an accurate summary of findings table (https://gradepro.org/).

Answer. Thank you for your observation. In Table 3, we evaluated the certainty of evidence through the GRADE methodology for each version of GDS-4 and GDS-5 for each cut-off point. As seen in the table, there are versions of GDS-4 or GDS-5 for each cut-point that have been evaluated by only one study. It is our understanding that the GRADE methodology can be performed on individual studies when only one study answers the question to be evaluated. This is indicated by GRADE in its article "GRADE guidelines: 12. Preparing Summary of Findings tables - binary outcomes" by giving examples of outcomes evaluated by only one randomized clinical trial in Table 2 and Table 3 (1).

1. Guyatt GH, Oxman AD, Santesso N, Helfand M, Vist G, Kunz R, Brozek J, Norris S, Meerpohl J, Djulbegovic B, Alonso-Coello P, Post PN, Busse JW, Glasziou P, Christensen R, Schünemann HJ. GRADE guidelines: 12. Preparing summary of findings tables-binary outcomes. J Clin Epidemiol. 2013;66(2):158-72. doi: 10.1016/j.jclinepi.2012.01.012.

---

## [Decision Letter · Decision Letter 1]

26 May 2021

PONE-D-21-03547R1

Accuracy of the Geriatric Depression Scale (GDS)-4 and GDS-5 for the screening of depression among older adults: a systematic review and meta-analysis

PLOS ONE

Dear Alvaro Taype-Rondan,

Thank you for submitting your manuscript to PLOS ONE. After careful consideration, we feel that it has merit but does not fully meet PLOS ONE’s publication criteria as it currently stands. Therefore, we invite you to submit a revised version of the manuscript that addresses the points raised during the review process.

Please respond to review comments about clarifying what constitutes the overall meta-analyses versus investigations of heterogeneity in the abstract and main text (any versions vs similar versions at different common cutoff points?). In addition, do ensure that the conclusions of the review are in line with the stated objectives in both the abstract and main text.

We look forward to receiving your revised manuscript.

Kind regards,

Eleanor Ochodo, M.D., PhD

Academic Editor

PLOS ONE

Journal Requirements:

Reviewers' comments:

Reviewer's Responses to Questions

**Comments to the Author**

1. If the authors have adequately addressed your comments raised in a previous round of review and you feel that this manuscript is now acceptable for publication, you may indicate that here to bypass the “Comments to the Author” section, enter your conflict of interest statement in the “Confidential to Editor” section, and submit your "Accept" recommendation.

Reviewer #1: All comments have been addressed

Reviewer #2: (No Response)

2. Is the manuscript technically sound, and do the data support the conclusions?

Reviewer #1: Yes

Reviewer #2: Yes

3. Has the statistical analysis been performed appropriately and rigorously? 

Reviewer #1: Yes

Reviewer #2: No

4. Have the authors made all data underlying the findings in their manuscript fully available?

Reviewer #1: Yes

Reviewer #2: Yes

5. Is the manuscript presented in an intelligible fashion and written in standard English?

Reviewer #1: Yes

Reviewer #2: Yes

6. Review Comments to the Author

Reviewer #1: (No Response)

Reviewer #2: The authors have responded satisfactorily to the previous comments except those about thresholds.

It is still unclear what the overall meta-analyses entails versus investigations of heterogeneity. Overall meta-analysis (any GDS-4 version or GDS-5 version separately at different common cutoffs? ) vs heterogeneity (same GDS version at different common cutoffs). Since different cut-offs and versions have been used it is important to be clear from the outset what constitutes the overall meta-analysis vs heterogeneity.

ABSTRACT:

Methods section:

" We conducted sensitivity and specificity meta-analyses of those studies that used.....". Please revise to we conducted meta-analyses of the sensitivity and specificity of those studies that used......

"We performed a bivariate random-effects meta-analysis to calculate the pooled sensitivity and specificity with their 95% confidence intervals (95% CI). " Please be very explicit what this overall meta-analyses included. For example one could rephrase as follows "we performed a bivariate random-effects meta-analysis to estimate the pooled sensitivity and specificity with their 95% confidence intervals (95% CI) at each reported common cutoff. For the overall meta-analyses, any GDS-4 version or GDS-5 version separately, by each cut-off and for investigations of heterogeneity, across similar GDS versions by each cutoff".

Results

Being very explicit about the cutoff and versions will help one understand the results better. For example, the first set of reported results is about all versions of GDS-4 and GDS-5 separately at cutoff>2. This implies the overall meta-analyses at cutoff 2?? and subsequent cutoffs??

Conclusion:

This conclusion does not accurately reflect the aim of the review which is to estimate accuracy. For example this could be reworded to " We found several GDS-4 and GDS-5 versions that showed great heterogeneity in estimates of sensitivity and specificity, mostly with a low or very low certainty of the evidence"...........

MAIN TEXT

Statistical analyses section page 7

The overall meta-analyses vs heterogeneity is unclear. For example please see comparisons below:

Lines 133-135

We conducted meta-analyses of the sensitivity and specificity of GDS-4 and GDS-5 versions whenever studies fulfilled the following condition: 1) There was more than one study that compared the same version of GDS-4 or GDS-5 and used the same cut-off point.

Lines 143-144

In addition, we meta-analyzed all the included studies that assessed any GDS-4 version, and all the studies that assessed any GDS-5 version, by each cut-off.

The first statements about meta-analyses (lines 133-135) seem similar to investigations of heterogeneity (lines 145-146).

7. PLOS authors have the option to publish the peer review history of their article (what does this mean?). If published, this will include your full peer review and any attached files.

Reviewer #1: **Yes: **Roshana Shrestha

Reviewer #2: No

---

## [Author Response · Author response to Decision Letter 1]

11 Jun 2021

Journal Requirements:

Answer. Thank you for your observation. We have reviewed the list of references and made changes to the reference style to make it conform to the journal's requirements. We found that Cheng 2004 (1) and Tsoi 2017 (2) have been corrected. We added their “Erratum” information in the references (Lines 383-384 and Line 420).

The list of references is complete, and no retracted studies have been cited.

According to the corrected information from Cheng 2004 (1), we have changed in the column “Reference standard” of Table 1: “Major depressive disorders, dysthymia, adjustment disorder with depressed mood, dementia with depression, bipolar disorder, depressive episode assessed by DSM-III” to “Major depressive disorder, dysthymia, depressive disorder not otherwise specified, adjustment disorder with depressed mood, dementia with depression assessed by DSM-III”. This change was also made in the Supplementary File S3_Table. No other changes to the manuscript were required.

1. Cheng ST, Chan AC. A brief version of the geriatric depression scale for the chinese. Psychol Assess. 2004 Jun;16(2):182-6. doi: 10.1037/1040-3590.16.2.182. Erratum in: Psychol Assess. 2006 Mar;18(1):48. PMID: 15222814.

2. Tsoi KK, Chan JY, Hirai HW, Wong SY. Comparison of diagnostic performance of Two-Question Screen and 15 depression screening instruments for older adults: systematic review and meta-analysis. Br J Psychiatry. 2017;210(4):255-60. doi: 10.1192/bjp.bp.116.186932. Erratum in: Br J Psychiatry. 2017;211(2):120. pmid: 28209592.

Reviewers' comments:

R2.1: ABSTRACT 

Methods section:

" We conducted sensitivity and specificity meta-analyses of those studies that used.....". Please revise to we conducted meta-analyses of the sensitivity and specificity of those studies that used......

Answer. Thank you for your recommendation. We have changed in Lines 31-32: “We conducted sensitivity and specificity meta-analyses of those studies that used…” to “We conducted meta-analyses of the sensitivity and specificity of those studies that used…”.

R2.2: "We performed a bivariate random-effects meta-analysis to calculate the pooled sensitivity and specificity with their 95% confidence intervals (95% CI). " Please be very explicit what this overall meta-analyses included. For example one could rephrase as follows "we performed a bivariate random-effects meta-analysis to estimate the pooled sensitivity and specificity with their 95% confidence intervals (95% CI) at each reported common cutoff. For the overall meta-analyses, any GDS-4 version or GDS-5 version separately, by each cut-off and for investigations of heterogeneity, across similar GDS versions by each cutoff".

Answer. Thank you for your observation. We have changed in Lines 34-39: “We performed a bivariate random-effects meta-analysis to calculate the pooled sensitivity and specificity with their 95% confidence intervals (95% CI)” to “We performed bivariate random-effects meta-analyses to calculate the pooled sensitivity and specificity with their 95% confidence intervals (95% CI) at each reported common cut-off. For the overall meta-analyses, we evaluated each GDS-4 version or GDS-5 version separately by each cut-off, and for investigations of heterogeneity, we assessed altogether across similar GDS versions by each cut-off”.

R2.3: Results

Being very explicit about the cutoff and versions will help one understand the results better. For example, the first set of reported results is about all versions of GDS-4 and GDS-5 separately at cutoff>2. This implies the overall meta-analyses at cutoff 2?? and subsequent cutoffs??

Answer. Thank you for your observation. The sentence implies the overall meta-analyses at cut-off 2. It does not imply subsequent cut-offs. We have changed “at a cut-off ≥ 2” to “at a cut-off 2” for better understanding.

In addition, throughout the manuscript, the “≥” sign was eliminated when referring to cut-off points. For example, we have changed “cut-off ≥ 1” to “cut-off 1”.

R2.4: Conclusion:

This conclusion does not accurately reflect the aim of the review which is to estimate accuracy. For example this could be reworded to " We found several GDS-4 and GDS-5 versions that showed great heterogeneity in estimates of sensitivity and specificity, mostly with a low or very low certainty of the evidence"...........

Answer. Thank you for your observation. We have changed in Line 51: “We found several GDS-4 and GDS-5 versions that showed great heterogeneity, mostly with a low or very low certainty of the evidence” to “We found several GDS-4 and GDS-5 versions that showed great heterogeneity in estimates of sensitivity and specificity, mostly with a low or very low certainty of the evidence”

R2.5: MAIN TEXT

Statistical analyses section page 7

The overall meta-analyses vs heterogeneity is unclear. For example please see comparisons below:

Lines 133-135

We conducted meta-analyses of the sensitivity and specificity of GDS-4 and GDS-5 versions whenever studies fulfilled the following condition: 1) There was more than one study that compared the same version of GDS-4 or GDS-5 and used the same cut-off point.

Lines 143-144

In addition, we meta-analyzed all the included studies that assessed any GDS-4 version, and all the studies that assessed any GDS-5 version, by each cut-off.

Answer. Thank you for your observation. For the overall meta-analyses, we evaluated each GDS-4 version or GDS-5 version separately by each cut-off. Regarding investigations of heterogeneity, we assessed altogether across any GDS-4 version or GDS-5 version by each cut-off. 

We have changed Lines 133-135 to “We conducted meta-analyses of the sensitivity and specificity of each of the GDS-4 and GDS-5 versions whenever studies fulfilled the following condition: 1) There was more than one study that compared the same version of GDS-4 or GDS-5 at the same cut-off point”.

We have changed Lines 143-144 to “In addition, we meta-analyzed altogether the results of the included studies that assessed the same cut-off point of any GDS-4 version. Likewise, we meta-analyzed altogether the results of the included studies that assessed the same cut-off point of any GDS-5 version”.

---

## [Editor Report · Decision Letter 2]

16 Jun 2021

Accuracy of the Geriatric Depression Scale (GDS)-4 and GDS-5 for the screening of depression among older adults: a systematic review and meta-analysis

PONE-D-21-03547R2

Dear Alvaro Taype-Rondan,

We’re pleased to inform you that your manuscript has been judged scientifically suitable for publication and will be formally accepted for publication once it meets all outstanding technical requirements.

Kind regards,

Eleanor Ochodo

Academic Editor

PLOS ONE

---

## [Editor Report · Acceptance letter]

21 Jun 2021

PONE-D-21-03547R2 

Accuracy of the Geriatric Depression Scale (GDS)-4 and GDS-5 for the screening of depression among older adults: a systematic review and meta-analysis 

Dear Dr. Taype-Rondan:

I'm pleased to inform you that your manuscript has been deemed suitable for publication in PLOS ONE. Congratulations! Your manuscript is now with our production department. 

Kind regards, 

on behalf of

Dr. Eleanor Ochodo 

Academic Editor

PLOS ONE